# Central Nervous System Disorders of Marine Mammals: Models for Human Disease?

**DOI:** 10.3390/pathogens13080684

**Published:** 2024-08-14

**Authors:** Giovanni Di Guardo

**Affiliations:** Former Professor of General Pathology and Veterinary Pathophysiology, Veterinary Medical Faculty, University of Teramo, Località Piano d’Accio, 64100 Teramo, Italy; gdiguardo@unite.it

**Keywords:** central nervous system disorders, marine mammals, Alzheimer’s disease, subacute sclerosing panencephalitis, *measles virus*, *Cetacean Morbillivirus*, *A(H5N1) avian influenza virus*, *Brucella ceti*

## Abstract

This article deals with Central Nervous System (CNS) disorders of marine mammals as putative neuropathology and neuropathogenesis models for their human and, to some extent, their animal “counterparts” in a dual “One Health” and “Translational Medicine” perspective. Within this challenging context, special emphasis is placed upon Alzheimer’s disease (AD), provided that AD-like pathological changes have been reported in the brain tissue of stranded cetacean specimens belonging to different Odontocete species. Further examples of potential comparative pathology interest are represented by viral infections and, in particular, by “Subacute Sclerosing Panencephalitis” (SSPE), a rare neurologic *sequela* in patients infected with *Measles virus* (MeV). Indeed, *Cetacean morbillivirus* (CeMV)-infected striped dolphins (*Stenella coeruleoalba*) may also develop a “brain-only” form of CeMV infection, sharing neuropathological similarities with SSPE. Within this framework, the global threat of the *A(H5N1) avian influenza virus* is another major concern issue, with a severe meningoencephalitis occurring in affected pinnipeds and cetaceans, similarly to what is seen in human beings. Finally, the role of *Brucella ceti*-infected, neurobrucellosis-affected cetaceans as putative neuropathology and neuropathogenesis models for their human disease counterparts is also analyzed and discussed. Notwithstanding the above, much more work is needed before drawing the conclusion marine mammal CNS disorders mirror their human “analogues”.

A relevant and emerging research area in marine mammal neuroscience is that of Central Nervous System (CNS) disorders as potential neuropathology and neuropathogenesis models for their human and animal disease “counterparts”, within a dual “One Health” and “Translational Medicine” perspective.

This holds especially true for Alzheimer’s disease (AD) and “Subacute Sclerosing Panencephalitis” (SSPE), two human central neuropathies, as well as for a canine CNS disorder, popularly known as “Old Dog Encephalitis” (ODE).

Indeed, whereas AD represents the most frequent form of dementia worldwide, no animal models faithfully recapitulating the whole spectrum of its key neuropathological and neuropathogenetic features appear to exist [1], although elderly dogs with cognitive dysfunction syndrome may show cognitive deficits typical of AD in human beings [2]. Within this framework, AD-like lesions consistent with amyloid-β (Aβ) extra-neuronal plaques and hyperphosphorylated tau protein neurofibrillary tangles have been reported in the brain tissue of striped dolphins (*Stenella coeruleoalba*) and bottlenose dolphins (*Tursiops truncatus*) found stranded along the Spanish coastline [3]. Furthermore, AD-related immunohistopathological changes have been described in stranded beaked whales (*Ziphiidae*), with these findings having been linked to cerebral hypoxia [4], while an AD-like central neuropathy has also been reported in three oceanic odontocete species, namely, bottlenose dolphin, white-beaked dolphin (*Lagenorhynchus albirostris*) and long-finned pilot whale (*Globicephala melas*) [5]. Beta-N-methylamino-L-alanine (BMAA), a cyanobacterial neurotoxin accumulating inside marine trophic chains, has been deemed responsible for Aβ plaques and dystrophic neurites in bottlenose and common dolphins (*Delphinus delphis*) stranded along the Atlantic USA seaboard [6], with a TAR DNA-binding protein 43 (TDP-43) proteinopathy and AD-like brain lesions having been additionally reported in a BMAA-exposed harbor porpoise (*Phocoena phocoena*) [7].

Of course, this does not necessarily imply that cetaceans may develop a true AD encephalopathy as we know it in people. Therefore, a much deeper genetic, biomolecular and phenotypical characterization of their CNS disorder is needed before drawing the conclusion that it mirrors (or does not mirror) human AD. In this respect, given the pivotal roles played by the cellular prion protein (PrP^C^) as a high-affinity neuronal receptor for Aβ oligomers and in neurotoxic Aβ oligomer-induced synaptic dysfunction in AD [8], a comparative study on brain PrP^C^ expression in cetaceans with AD-like alterations could provide valuable information on the pathogenetic links of their encephalopathy, if any, with human AD [9].

Similarly to patients infected with *Measles virus* (MeV) and affected by Subacute Sclerosing Panencephalitis (SSPE), and to canines infected with *Canine distemper virus* (CDV) and affected by Old Dog Encephalitis (ODE), a peculiar “brain-only” form of *Cetacean morbillivirus* (CeMV) infection may also occur in cetaceans [10,11,12], with SSPE-like lesions having been reported in 25% of striped dolphins found stranded along the Mediterranean coast of Spain after the 2006–2007 CeMV epidemic [13]. Indeed, throughout the last 35 years, no less than 10 morbilliviral epidemics have affected free-ranging cetacean populations from different areas of the planet, with striped dolphins being the main “target” of the four outbreaks that occurred in the Western Mediterranean Sea between 1990 and 2013, and with bottlenose dolphins being the species most consistently affected by the four unusual mortality events that occurred between 1982 and 2014 along the Atlantic USA seaboard [11].

The main feature of this “brain-only” form of cetacean neuropathy, sharing morpho-pathological traits with human SSPE and canine ODE, is that CeMV antigens and/or genomes are found only at brain level [12,13]. Despite the identification of the CeMV-infected neuronal and glial cell populations from the cerebral tissue of striped dolphins with SSPE-like lesions [14], the cell receptor(s) and the viral determinants allowing viral persistence and spread throughout the brain are yet to be defined, with this knowledge gap also applying to SSPE-affected humans [1,11,15,16]. In this respect, human neurons do not express the two main morbilliviral host–cell receptors, namely, the signaling lymphocytic activation molecule (SLAM, *alias* CD150) and nectin-4, *alias* the poliovirus-receptor-like-4 (PVRL4), respectively specifying the well-known lymphotropism and epitheliotropism exhibited by MeV as well as by all the other *Morbillivirus* genus members [16]. Strikingly enough, however, the nectin-4 receptor is expressed by CNS cells in dogs, being also involved in CDV neurovirulence [17]. Furthermore, mutations of the phosphoprotein (P), matrix (M) and fusion (F) protein genes have been reported to promote MeV spread in infected human brains, thus correlating with SSPE development [15,18,19]. Within this complex framework, an intriguing interaction pathway between MeV haemagglutinin (H) and host–cell adhesion molecules 1 (CADM1) and 2 (CADM2) has been recently described in neurons and other brain cells lacking SLAM and nectin-4; this mechanism triggers, in fact, hyperfusogenic F protein-mediated membrane fusion and, consequently, cell-to-cell trans-synaptic viral spread [20]. Of note, only short-stalk isoforms of CADM1 and CADM2, which are predominantly expressed in the brain, have been reported to induce hyperfusogenic F protein-mediated membrane fusion [21].

Since studies of this kind are yet to be performed in CeMV-infected cetaceans, caution should be taken before crediting striped dolphins as a comparative pathology model for human SSPE and SSPE-like neuropathies. To this aim, in-depth analyses of P, M, F and H gene mutations in CeMV isolates from cetaceans carrying SSPE-like lesions could shed light on how the virus spreads and persists in their brain tissue [1,16].

Still with reference to viral infections, the prominent neurotropism and neuropathogenicity exhibited (also) in marine mammals by the highly pathogenic avian influenza (HPAI) *A(H5N1) virus* (clade 2.3.4.4b) are of significant concern to their health and conservation. Indeed, mass mortality outbreaks have been recently caused by such pathogen among South American sea lions (*Otaria flavescens*), along with cases of fatal encephalitis in cetaceans from Swedish and Atlantic USA waters, as well as in a polar bear from Alaska (*Ursus maritimus*) [22,23,24,25,26]. Notably, all affected marine mammal species are included into the *International Union for the Conservation of Nature* (IUCN) *Red List of Threatened Species*, with the polar bear being currently classified as “Vulnerable” (VU) on the basis of a projected reduction in global population size due to the loss of sea ice habitat (https://www.iucn.org/sites/default/files/2023-09/2022-iucn-ssc-polar-bear-sg-report_publication.pdf, accessed on 27 July 2024). Alongside the genetic mutations allowing *A(H5N1) avian influenza virus* to infect and adapt to a progressively expanding number of domestic and wild avian, terrestrial and aquatic mammalian species, a key issue is that of host–pathogen interaction dynamics, with special emphasis on the species- and the virus-related factors driving the neurotropism and neuropathogenicity displayed in both pinniped and cetacean hosts.

Within such a challenging and intriguing *scenario*, the neuropathogenesis of the CNS lesions found in *Brucella ceti*-infected striped dolphins is of additional interest from a One Health and Translational Medicine perspective. Indeed, *B. ceti*—a zoonotic bacterial agent infecting striped dolphins and other cetaceans across the globe—may also cause a fatal neurobrucellosis disease condition in the former species [27]. Getting inspiration from a study reporting PrP^C^ as the host–cell receptor for *B. abortus* on murine macrophages [28], PrP^C^ expression was analyzed in the brain tissue from *B. ceti*-infected, neurobrucellosis-affected striped dolphins beached along the coasts of Italy and the Canary Islands [29]. A clear-cut PrP^C^ immunoreactivity, albeit of different intensity, was found in all *B. ceti*-infected, neurobrucellosis-affected striped dolphins under investigation, with a number of “intrinsic” and “extrinsic” factors—complementary or alternative to *B. ceti* infection/neurobrucellosis—reasonably justifying, however, the aforementioned differences in the intensity of brain PrP^C^ expression [29]. Therefore, much more work is needed in order to properly assess the role of PrP^C^, if any, as a host–cell receptor for *B. ceti* in striped dolphins. Still of interest from a dual One Health and Translational Medicine perspective, a neuroinflammatory lesion pattern showing similarities with that seen in human neurobrucellosis has also been reported in *B. ceti*-infected, neurobrucellosis-affected Odontocete cetaceans [27,30], with macrophages/microglial cells, T-lymphocytes and B-cells infiltrating in equal proportion the leptomeninges, ependyma and/or choroid plexuses of *B. ceti*-infected striped dolphins [30].

In conclusion, marine mammal diseases, with special emphasis on cetacean CNS disorders, are important from a “One Health” and a “Translational Medicine” perspective, thus acting as putative “mirrors” for their human “counterparts”. This should be properly brought to the attention of marine mammal pathologists and, more in general, of the biomedical community, while also paving the way for an increased interest into the neurological disease conditions affecting these iconic sea creatures. Within such a challenging and intriguing context, the potential impact of CNS disorders on marine mammals’ health and conservation should be carefully assessed through in-depth *post mortem* and ancillary investigations on stranded individuals, including ad hoc protocols allowing us to characterize the differential expression of host genes driving disease susceptibility/resistance [1].

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
