# Peer review of "Central Nervous System Disorders of Marine Mammals: Models for Human Disease?"

_pathogens, 2024, doi:10.3390/pathogens13080684_

Round 1

Reviewer 1 Report (Previous Reviewer 1)

Comments and Suggestions for Authors

The revised version of the manuscript has addressed all the changes suggested and now can be considered for full acceptance. 

Author Response

I would like to warmely thank this Reviewer for her/his extremely valuable and precious comments, remarks and suggestions, which have greatly contributed - alongside those kindly provided by the Academic Editor - to enhance the overall quality of this Perspective article.

Reviewer 2 Report (Previous Reviewer 2)

Comments and Suggestions for Authors

Dear author,

thanks for the changes. I think this article will guide the next generation of marine mammal pathologist in the rigth way to discover very relevanlt findings, that may be extrapolated to human and global health. 

Regards. 

Author Response

I would like to warmely thank this Reviewer for her/his extremely valuable and precious comments, remarks and suggestions, which have greatly contributed - alongside those kindly provided by the Academic Editor - to enhance the overall quality of this Perspective article.

Reviewer 3 Report (Previous Reviewer 3)

Comments and Suggestions for Authors

the revised ms meets the requests made by this reviewer

Author Response

I would like to warmely thank this Reviewer for her/his extremely valuable and precious comments, remarks and suggestions, which have greatly contributed - alongside those kindly provided by the Academic Editor - to enhance the overall quality of this Perspective article.

This manuscript is a resubmission of an earlier submission. The following is a list of the peer review reports and author responses from that submission.

Round 1

Reviewer 1 Report

Comments and Suggestions for Authors

The paper submitted by the author provides information about CNS disorders in marine mammals and their similarities with those disorders identified in humans. Before acceptance, some changes should be done. These suggestions are listed below.

In the abstract, more of the examples are related to viral infections. Thus, begin the paragraph with a sentence that collated all of them.

Lines 22-23, these are a little bit redundant and already explained before, please delete and redone.

As you stated at the end of the abstract section, more studies should be done to stablish marine mammals as models of CNS human diseases. Then, I suggest to modify the title of the paper.

In the keywords, please change aquatic mammals by marine mammals. Lines 36-37,  here you should include cognitive dysfunction syndrome (CDS). It is a common age-related disease in dogs that affects the brain, causing deterioration similar to Alzheimer's disease in humans.

Line 79, please add some more about the morbiliiviral receptor.

Line 104, please check It’s status at the IUCN list and add the information in the sentence.

Finally, the author should take special attention to the self-citation since nine out of thirty references belong to their own publications.

Reviewer 2 Report

Comments and Suggestions for Authors

Line 103. The link is not working. Also a scientific reference would be better. The case of the polar bear was Influenza? if not related maybe is not needed. 

Lines 105-109. This pharagraph is long and difficult to follow. I would recommend to restructure this part.

The following article also can be cited in the comparisong of the inflammatory patter of neurobrucellosis in humans and dolphins:  

Sierra E, Fernández A, Felipe-Jiménez I, Zucca D, Díaz-Delgado J, Puig-Lozano R, Câmara N, Consoli F, Díaz-Santana P, Suárez-Santana C, Arbelo M. Histopathological Differential Diagnosis of Meningoencephalitis in Cetaceans: Morbillivirus, Herpesvirus, Toxoplasma gondii, Brucella sp., and Nasitrema sp. Front Vet Sci. 2020 Sep 30;7:650. doi: 10.3389/fvets.2020.00650

Reviewer 3 Report

Comments and Suggestions for Authors

This ms is a timely perspective of encephalitis in marine mammals. The document is interesting however the Engliosh grammar needs to be reviewed in some parts. 

A suggestion is to add a table or paragraph with the most recent outbreaks or reported pathogens in the marine environment to enhance the scientific content of the ms.

Comments on the Quality of English Language

English grammar needs to be improved in some sections.

Author’s reply

I would like to thank the Editor for her/his relevant comments, remarks and criticisms, based upon the confidential comments provided by Reviewers 1 and 3. In this respect, let me kindly emphasize that the present Perspective and my previous Commentary published in 2023 in Veterinary Pathology, albeit dealing with the "role of marine mammals' CNS disorders as putative models for their human and animal disease counterparts", are still quite dissimilar from each other in their contents. Just to make a convincing example about the number of remarkable and substantial differences between the two manuscripts, the A(H5N1) avian influenza virus neurotropism and neuropathogenicity to marine mammals are not even mentioned in the Vet Pathol Commentary. The latter contribution, in fact, is exclusively focused on Alzheimer's disease- and Subacute Sclerosing Panencephalitis (SSPE)-like pathological changes in marine mammals, with no mention at all of the neuropathological and neuropathogenetic features of Brucella ceti infection in neurobrucellosis-affected cetaceans in a dual One Health and Translational Medicine perspective. Conversely, these are highlighted in the current Perspective article, the revision of which (on the basis of the highly valuable and precious comments, remarks and suggestions kindly made by you as well as by the three Reviewers) renders/makes it now even more distant from the contents of the aforementioned Commentary previous published by myself in Veterinary Pathology.

Reviewer 1

The paper submitted by the author provides information about CNS disorders in marine mammals and their similarities with those disorders identified in humans. Before acceptance, some changes should be done. These suggestions are listed below.

In the abstract, more of the examples are related to viral infections. Thus, begin the paragraph with a sentence that collated all of them.

Reply: I would like to thank the Reviewer for this useful suggestion on her/his behalf, which I have strictly followed, thus adding an ad hoc sentence at the beginning of the concerned paragraph.

Lines 22-23, these are a little bit redundant and already explained before, please delete and redone.

Reply: Done.

As you stated at the end of the abstract section, more studies should be done to stablish marine mammals as models of CNS human diseases. Then, I suggest to modify the title of the paper.

Reply: The title has been modified, as kindly suggested by the Reviewer.

In the keywords, please change aquatic mammals by marine mammals. Lines 36-37, here you should include cognitive dysfunction syndrome (CDS). It is a common age-related disease in dogs that affects the brain, causing deterioration similar to Alzheimer's disease in humans.

Reply: I would like to thank the Reviewer for her/his valuable and precious comments, remarks and suggestions, which have been duly followed in and applied to the present manuscript’s revision.

Line 79, please add some more about the morbiliiviral receptor.

Reply: Thank you very much for this useful suggestion! Some more details on the morbilliviral receptor have now been incorporated into the current manuscript’s revision.

Line 104, please check It’s status at the IUCN list and add the information in the sentence.

Reply: Done. Thank you very much for this useful suggestion on your behalf!

Finally, the author should take special attention to the self-citation since nine out of thirty references belong to their own publications.

Reply: Thank you very much also for this useful and precious comment and remark, which I have srictly followed, thereby deleting a number of self-citations from the text.

Reviewer 2

Line 103. The link is not working. Also a scientific reference would be better. The case of the polar bear was Influenza? if not related maybe is not needed.

Reply: I would like to thank the Reviewer for her/his valuable and precious comments, remarks and suggestions, which have been duly followed in and applied to the present manuscript’s revision.

Indeed, a case of A(H5N1) avian influenza virus infection was ascertained at the beginning of this year in an Alaskan polar bear (Ursus maritimus). In this respect, an ad hoc reference has been also added to the revised manuscript’s text, as usefully suggested by the Reviewer.

Lines 105-109. This pharagraph is long and difficult to follow. I would recommend to restructure this part.

Reply: The paragraph has been re-written, thus improving its clarity, as usefully suggested by the Reviewer.

The following article also can be cited in the comparisong of the inflammatory patter of neurobrucellosis in humans and dolphins:

Sierra E, Fernández A, Felipe-Jiménez I, Zucca D, Díaz-Delgado J, Puig-Lozano R, Câmara N, Consoli F, Díaz-Santana P, Suárez-Santana C, Arbelo M. Histopathological Differential Diagnosis of Meningoencephalitis in Cetaceans: Morbillivirus, Herpesvirus, Toxoplasma gondii, Brucella sp., and Nasitrema sp. Front Vet Sci. 2020 Sep 30;7:650. doi: 10.3389/fvets.2020.00650

Reply: Thank you very much for this kind and useful suggestion! This relevant and interesting bibliographic reference has now been properly included into the present manuscript’s revision.

Reviewer 3

This ms is a timely perspective of encephalitis in marine mammals. The document is interesting however the English grammar needs to be reviewed in some parts.

Reply: I would like to thank the Reviewer for her/his valuable and precious comments, remarks and suggestions, to which adequate attention has been paid by myself in revising this manuscript.

A suggestion is to add a table or paragraph with the most recent outbreaks or reported pathogens in the marine environment to enhance the scientific content of the ms.

Reply: Thank you very much for this useful and precious suggestion! An ad hoc paragraph summarizing the CeMV infection/disease outbreaks worldwide has now been incorporated into the current manuscript’s revision. Furthermore, some more details concerning the recent A(H5N1) avian influenza outbreaks among marine mammals have also been included, alongside some more details concerning Brucella ceti infection.

English grammar needs to be improved in some sections.

Reply: The English grammar has been duly revised and improved all throughout the manuscript, thanks to the useful and precious support of a native English speaking person.